# Drone $CO_2$ Measurements During the Tajogaite Volcanic Eruption

John Ericksen[1], Tobias P. Fischer[2], G. Matthew Fricke[1], Scott Nowicki[2], Nemesio M. Pérez[3], Pedro Hernández Pérez[3], Eleazar Padrón González[3], and Melanie E. Moses[1,4]

[1]Computer Science Department, University of New Mexico, Albuquerque, New Mexico, USA
[2]Department of Earth and Planetary Sciences, University of New Mexico, Albuquerque, New Mexico, USA
[3]Instituto Volcanológico de Canarias (INVOLCAN), Tenerife, Islas Canarias, Spain
[4]Santa Fe Institute, Santa Fe, New Mexico, USA

**Correspondence:** Tobias P. Fischer (fischer@unm.edu), G. Matthew Fricke (mfricke@unm.edu)

**Abstract.** We report in-plume carbon dioxide ($CO_2$) concentrations and carbon isotope ratios during the 2021 eruption of Tajogaite Volcano, La Palma Island, Spain. $CO_2$ measurements inform our understanding of volcanic contributions to the global climate carbon cycle and the role of $CO_2$ in eruptions. Traditional ground-based methods of $CO_2$ collection are difficult and dangerous and as a result only about 5% of volcanoes have been directly surveyed. We demonstrate that UAS surveys allow for fast and relatively safe measurements. Using $CO_2$ concentration profiles we estimate the total flux during several measurements in November 2021 to be $1.76 \pm 0.20 \times 10^3$ to $2.23 \pm 0.26 \times 10^4 \, \mathrm{t \, day^{-1}}$. Carbon isotope ratios of plume $CO_2$ indicate a deep magmatic source, consistent with the intensity of the eruption. Our work demonstrates the feasibility of UAS for $CO_2$ surveys during active volcanic eruptions, particularly for deriving rapid emission estimates.

## 1 Introduction

Measurements of volcanic $CO_2$ emissions during eruptions are critical for understanding magma and eruption dynamics. $CO_2$ is a significant greenhouse gas (Arrhenius, 1896) and making measurement of $CO_2$ emissions is important for climate science. $CO_2$ gas is second only to water vapor in abundance in volcanic emissions (Giggenbach, 1996). Despite the significance and abundance of $CO_2$ in the Earth System in general and in magmatic systems in particular, measuring the emission rates of this gas from volcanic craters, diffuse sources, and low-level hydrothermal sites has remained a major challenge (Fischer and Aiuppa, 2020). As a result, detailed $CO_2$ surveys have been conducted at just 5% of volcanoes (Fischer et al., 2019).

The main contributions of this work are that, for the first time, we estimate $CO_2$ flux using direct in-plume $CO_2$ measurements rather than using in-plume $CO_2$ to $SO_2$ ratios combined with separately measured $SO_2$ emissions. The second major contribution is that we perform in-situ gas sample-return during a major volcanic eruption for carbon isotope measurements. We use the Dragonfly Unpiloted Aerial System (UAS) (Ericksen et al., 2022) to gather samples directly from the eruption plume (Figure 1). The UAS transects the plume and employs an onboard infrared (IR) sensor to continuously obtain concentration readings. These readings are then used to estimate a 2D isotropic Gaussian concentration model. In-plume wind velocity measurements in combination with the plume model allow us to estimate $CO_2$ flux. While our technique has similarities to the 'ladder traverse' technique utilizing large in-situ sensing equipment mounted on a piloted fixed-wing aircraft (Werner et al.,

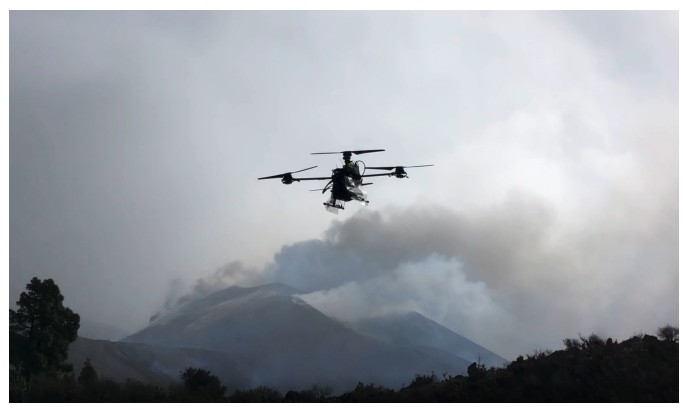

**Figure 1. A Dragonfly UAS returning from a CO₂ sample mission during the November 2021 eruption of Tajogaite volcano.** The large volcanic ash plume is visible in the background and contains an invisible $CO_2$ plume, which was the mapping target of this drone.

2013), it has the obvious advantages of being much less costly, logistically less challenging, and less hazardous. Since our approach extrapolates the shape of the plume it requires far fewer plume transects. Crucially, the Dragonfly UAS does not use a combustion engine, which previous work has shown to contaminate $CO_2$ measurements and samples with jet-fuel derived organic carbon (Fischer and Lopez, 2016). The resulting plume $CO_2$ concentration profile is used to guide the UAS to a productive sample return location of maximum concentration. Carbon isotope analyses of the samples reveal information, such as $CO_2$ source, which is relevant to predicting the course of the eruption. We tested this technique during the 2021 Tajogaite volcanic eruption on La Palma Island, Spain, and compared the resulting flux estimates to the traditional ground-based $CO_2$ to $SO_2$ ratio method. As we demonstrate, UASs provide a method for obtaining in-plume gas samples, concentrations, and wind velocity measurements. Together these data allow isotope ratios to be determined and estimation of $CO_2$ flux, furthering our understanding of volcano dynamics during an eruption and allowing predictions of eruption intensity and duration. Our technique can be widely used at passively degassing and erupting volcanoes to obtain near-real-time $CO_2$ flux measurements to better constrain the global volcanic $CO_2$ budget, and assess volcanic activity.

## 1.1 Related Work

While global initiatives to directly determine $CO_2$ flux from biogenic sources, i.e. FLUXNET (Office of Science, US DOE, 2023) have advanced our understanding of the surface carbon cycle, estimates of volcanic flux are to a large extent obtained by combining $SO_2$ flux measurements with observed $CO_2$ to $SO_2$ ratios (Fischer and Aiuppa, 2020). This approach relies on two separate sets of measurements utilizing a ground-based or space-based remote sensing technique to determine the $SO_2$ concentration of the volcanic plume and a direct sampling or sensing technique to determine the $CO_2$ to $SO_2$ ratio. In almost all cases, these two separate sets of measurements are not made simultaneously and result in intrinsic uncertainties in $CO_2$ flux estimates (Burton et al., 2013). $CO_2$ surveys have been performed using satellite-based approaches, for example, Johnson et al.

(2020) performed $CO_2$ flux estimates of the 2018 Kilauea Volcano. Their work utilized the Orbiting Carbon Observatory -2
(OCO-2) to measure the $CO_2$ emissions from the 2018 Klīlauea eruption. A measurement of 77.1±41.6 kt/day was obtained
during the one day of observations where conditions enabled the collection of consistent high-quality data. Cloud coverage and
aerosol are the major inhibitors for obtaining consistent $CO_2$ data using OCO-2. In addition, the wind direction must be near
perpendicular to the satellite's orbit path and the measurements must be made down-wind from the plume. The OCO-2 16-day
repeat cycle currently makes this method impractical for frequent, high-rate $CO_2$ flux measurements from erupting volcanoes
and the only other successful volcanic $CO_2$ emission study was by Schwandner et al. (2017) of Yasur in Vanuatu. Therefore,
space-based $CO_2$ instruments require favorable atmospheric conditions and satellite positioning and are not yet feasible for
volcano monitoring (Schwandner et al., 2017).

The value of UAS surveys of volcanic emissions was recognized by Xi et al. (2016) who surveyed passive degassing $SO_2$ at
Turrialba volcano, Costa Rica and estimated $SO_2$ flux. Other investigators have used UAS to measure plume $SO_2$ and collect
plume trace gases (Rüdiger et al., 2018) or use miniDOAS systems mounted on UAV to obtain $SO_2$ fluxes (Stix et al., 2018).
Recently UAS have been used to collect gas samples and measure gas compositions volcanic plumes from passively degassing
volcanoes in remote regions (Liu et al., 2020; Galle et al., 2021) and during the 2023 eruption of Litli Hrútur, Iceland to obtain
information on $CO_2$ degassing and related carbon-isotope fractionation (Fischer et al., 2024)

Gerlach et al. (1997) and Werner et al. (2013) estimate plume $CO_2$ flux using the parsimonious assumption that plumes are
uniform. They use the mean value to estimate the flux whereas we use our observations in the field that support the hypothesis
that plumes can be well modeled by Gaussian distributions. Our work relies on the assumption that a Gaussian model of the
plume cross-section results in more accurate estimates of total flux.

Burton et al. (2023) surveyed emissions of the Tajogaite eruption in early October 2021. Their survey included $SO_2$ measure-
ments by UAV that were used to infer $CO_2$ concentrations. Our work in late November complements the Burton et. al. survey
by providing additional information on the evolution of the eruption and by using a different $CO_2$ flux estimation method that
employs direct $CO_2$ measurements rather than $CO_2/SO_2$ ratios. Our estimates of $CO_2$ flux taken a month later were lower
than those of Burton et. al.

## 1.2  Background

La Palma Island is in Spain's Canary archipelago (Schmincke, 1982). The northern sector of the island hosts the oldest subaerial
(on land) volcanism, characterized by repeated large lateral edifice collapses (Day et al., 1999; Acocella et al., 2015). Volcanism
resulted in the formation of Garafía and Taburiente and then moved southward to form Cumbre Vieja volcano, at the southern
part of the island. This southern system represents the last stage in the geological evolution of La Palma island, as volcanic
activity has taken place exclusively on that part of the island for the last 123 ka (Carracedo et al., 1998). The most recent
volcanic eruption of Cumbre Vieja is Tajogaite (2021) (Carracedo et al., 2001; Ward and Day, 2001), preceded by that of
Teneguía in 1971 (Fernández et al., 2021) and San Juan in 1940 (Fernández et al., 2021; Albert et al., 2016). At 14:10 UTC on
September 19, 2021 Tajogaite volcano erupted from a vent on the western side of La Palma Island, in the vicinity of the Llano
del Banco eruptive center of the San Juan eruption of 1949 (Instituto Geográfico Nacional, 2022). The eruption was forecast

using seismic, geodetic and geochemical techniques by Spanish researchers who alerted the civil protection officials several days before the start of the eruption (De Luca et al., 2022). The monitoring network of diffuse $CO_2$ emissions on La Palma detected magmatic $CO_2$ several months before the eruption (León et al., 2022; Rodríguez-Pérez et al., 2022). This monitoring activity took advantage of extensive previous work characterizing diffuse $CO_2$ emissions on La Palma. This work provided key insights into the dynamics of magmatic $CO_2$ degassing on the island (Padrón et al., 2015). The eruption itself began with an explosive phase that ejected ash to an altitude of 5 km, then transitioned to fire fountains, violent strombolian activity, and the production of highly fluid lava flows. Within 24 hours of the initial eruption a 3 km long lava flow was evident (Instituto Geográfico Nacional, 2022). The eruption lasted for more than 85 days and built a pyroclastic cone of about 225 m in height. Over the period of the eruption, the volcano showed dynamic and changing activity with new vents frequently opening on the active cone. These vents produced explosive and effusive eruptions of varying intensity (Castro and Feisel, 2022). Bulk tephra, matrix glass and glass inclusions have a basanitic-tephritic composition of 43 to 46 wt%.

Since the onset of the 2021 Tajogaite eruption on September 19, frequent measurements of $SO_2$ emission rates using miniDOAS traverses by car, ship, and helicopter were performed. Using this data a flux of over $5 \times 10^4 \, \mathrm{t \, day^{-1}}$ of $SO_2$ was estimated (Pérez et al., 2022). Daily monitoring of $SO_2$ gas emissions occurred before and throughout the eruption using TROPOMI data from the Sentinel 5P satellite (Copernicus $SO_2$ satellite monitoring, Smithsonian Institution's Global Volcanism Program 2021). The range of measured emissions rates depended upon wind direction and velocity, as well as eruptive style and activity. The measured $SO_2$ flux ranged from $3 \times 10^4$ to $5 \times 10^4 \, \mathrm{t \, day^{-1}}$ at the beginning of the eruption and a mean of $10^4 \, \mathrm{t \, day^{-1}}$ over the duration of the active eruption (Albertos et al., 2022). These $SO_2$ emission rates are likely different from $CO_2$, but provide the best available proxy for $CO_2$ emissions and are a useful point of comparison for our UAS-based flux estimates in addition to the measurements made by Burton et al. 2023 in October 2021 which range from $3.36 \times 10^4$ to $4.19 \times 10^4 \, \mathrm{t \, day^{-1}}$.

Additional gas monitoring techniques deployed during the eruption included stationary Multi-GAS and FTIR-based plume gas composition measurements as well as carbon isotope analyses of plume $CO_2$ in collaboration with the international volcanic gas community (Pérez et al., 2022).

## 2  Methods

Our aim was to measure plume $CO_2$ concentrations, calculate the resulting flux, and obtain isotope data from samples taken within the plume. To achieve these goals we utilized the Dragonfly UAS, with an approximate battery life of 50 min. This extended flight time enables long-distance transects to capture large plumes. $CO_2$ concentrations were measured by PP Systems SBA-5 IR sensor mounted on the Dragonfly with data transmitted to the pilot in real-time (Ericksen et al., 2022). Wind velocity and direction were derived from the ERA5 model of the European Centre for Medium-Range Weather Forecasts 10 m height wind velocities corresponding to the time of each flight (Liu et al., 2020). These measurements were independently validated using a hand-held anemometer and the UAS drift method (Liu et al., 2020; Galle et al., 2021). For the drift method, a Dragonfly was programmed to maintain its altitude but not its lateral position and allowed to drift with the plume. We used this estimate

of wind velocity within the plume with the highest $CO_2$ concentration (Plume B) to parameterize the flux estimation (Figure 2).

At the location with the highest measured $CO_2$ concentration, a timed trigger activated a small pump, and a plume gas sample was collected into a Tedlar bag (Figures 2 and 3). We also collected gas samples of the plume from the ground when

the wind direction was favorable and volcanic activity permitted. Ground-based plume samples were analyzed by Infrared Isotope Spectroscopy with a Delta Ray located at the INVOLCAN Volcano Observatory, La Palma, following the procedure described previously (Fischer and Lopez, 2016; Ilanko et al., 2019). The error bounds on the $\delta^{13}C$ measurements are less than 0.1‰ for all analyses.

We also placed a Multi-GAS instrument at an accessible and safe location about 1 km to the north of the crater. Data from

this instrument recorded $CO_2$ and $SO_2$ concentrations in the gas plume. The ratios were calculated using the Ratiocalc software and we report averages for each day of the experiment.

Crosswind transects were flown downwind of the eruption to encounter the plume. $CO_2$ was measured at 10 hz during flights across the plume at specified altitudes relative to launch. Each measurement was correlated to the latitude, longitude, altitude, and time of the UAS during flight, giving a $CO_2$ concentration cross-section of the plume.

We set the ambient background $CO_2$ to the value observed outside the plume for each flight. The actual measurements of ambient $CO_2$ were made well outside of the plume (up to 400 m away from the edge of the plume) and only vary from 415 to 430 ppm.

To estimate the total flux of the plume, we perform the following procedure.

1. Convert GPS coordinates into a linear distance in meters from the launch point. Each distance is normalised to the wind
direction perpendicular by multiplying it by $\cos(\text{heading}_{\text{uas}} - \text{heading}_{\text{wind}})$

2. Isolate the plume by setting an ambient $CO_2$ threshold and removing data points less than that threshold.

3. Fit a Gaussian curve to the data set as follows.

    (a) Calculate the mean, $\mu$, and standard deviation, $\sigma$, of the $CO_2$ across the transect.

    (b) Scale the two-dimensional Gaussian curve to fit the data by choosing a constant amplitude, $a$, using gradient descent
to minimize the squared difference between the model and plume sample data. We assume that the Gaussian shape is uniform in both $x$ and $y$ dimensions.

$$
\begin{aligned}
\text{GaussianModel2D}() &= a \frac{e^{-\frac{1}{2}(\frac{x-\mu_x}{\sigma_x})^2}}{\sigma_x\sqrt{2\pi}} \frac{e^{-\frac{1}{2}(\frac{y-\mu_y}{\sigma_y})^2}}{\sigma_y\sqrt{2\pi}} \\
&= a \frac{e^{-\frac{1}{2}(\frac{x-\mu}{\sigma})^2}}{\sigma\sqrt{2\pi}} \frac{e^{-\frac{1}{2}(\frac{0}{\sigma})^2}}{\sigma\sqrt{2\pi}} \qquad y=0, \mu_y=0, \sigma=\sigma_x, \sigma_y=\sigma, \mu=\mu_x \\
&= a \frac{e^{-\frac{1}{2}(\frac{x-\mu}{\sigma})^2}}{\sigma^2 2\pi}
\end{aligned}
$$

4. Integrate the two-dimensional Gaussian and multiply by the measured wind velocity, $v$, to obtain plume flux in $\mathrm{mg\,S^{-1}m^{-2}}$. Multiplying this again by the number of seconds in a day, and the number of $\mathrm{mg}$ in a ton gives the flux in $t\,day^{-1}$.

$$\int \mathrm{GaussianModel2D}() = a \int \frac{e^{-\frac{1}{2}\left(\frac{x-\mu}{\sigma}\right)^2}}{\sigma^2 2\pi} = a$$

$$\mathrm{flux}(a, v) = v\,a$$

Uncertainty in the flux calculation is given by the following root sum of squares method which combines the uncertainties in wind velocity $\epsilon_v$, wind direction $\epsilon_d$ sensor error $\epsilon_s$, and background $CO_2$ $\epsilon_b$. The total uncertainty, $\epsilon$, is calculated in accordance with the uncertainty estimation techniques described in Nassar et al. (2021); Lin et al. (2023); Nassar et al. (2017); Johnson et al. (2020):

$$\epsilon = \sqrt{\epsilon_v^2 + \epsilon_d^2 + \epsilon_s^2 + \epsilon_b^2}$$

## 3 Results

Flux estimates are derived from the 3 UAS transects that crossed plume A. These transects were collected on November 26th and 27th, 2021. Other transects shown in Figure 2 either did not intersect any plume or did not cross the entire plume. In the latter case this resulted in a poor fit to the Gaussian distribution, violating our assumption of normality. We also report carbon isotopes of plume $CO_2$, and flux estimates based on the Multi-GAS $CO_2/SO_2$ ratios.

**Table 1.** $CO_2$ data collected by UAS across plumes A and B during the Tajogaite eruption. * Indicates transect with samples collected into Tedlar bags and analyzed by Infrared Isotope Ratio Spectroscopy. † Indicates transects that encountered plume B, but the gas distribution did not meet our Gaussian fit assumptions, as indicated by the low $R^2$ value in comparison to the Gaussian amplitude. Thus we did not include plume B in our flux calculations.

| Date | Transect | Altitude | Wind $[\mathrm{m\,s^{-1}}@^\circ]$ | Max Con. [ppm] | Gaussian Fit Amplitude | $R^2$ | Flux $[\mathrm{t\,day^{-1}}]$ |
|---|---|---|---|---|---|---|---|
| 2021-11-26 | 2 Plume A | 200 m | 11.8 @ 68° | 501 | $8.95 \times 10^5$ | 0.93 | $1.76 \pm 0.20 \times 10^3$ |
| 2021-11-27 | 6 Plume A | 100 m | 12.2 @ 38° | 616 | $1.10 \times 10^7$ | 0.71 | $2.23 \pm 0.26 \times 10^4$ |
| 2021-11-27 | 7 Plume B † | 100 to 250 m | 12.2 @ 38° | 613 | $3.02 \times 10^6$ | 0.01 | $6.15 \pm 0.71 \times 10^3$ |
| 2021-11-27 | 8 Plume A | 300 m | 12.2 @ 38° | 577 | $2.81 \times 10^6$ | 0.75 | $5.71 \pm 0.66 \times 10^3$ |
| 2021-11-28 | 9 Plume B* † | 300 m | 11.3 @ 44° | 963 | $3.85 \times 10^7$ | 0.36 | $7.25 \pm 0.84 \times 10^4$ |

### 3.1 Plume Transect Wind Measurements

The calculated $CO_2$ flux for the 5 relevant transects with the corresponding wind velocities and directions are shown in Table 1 for transects across plume A and B. The wind velocity measured by UAS drift method was $10.7\ \mathrm{ms^{-1}}$. ERA5 modeled wind

**Table 2.** Measured $CO_2$ concentrations and $\delta^{13}C$ from ground and UAS.

| Date | $CO_2$ [ppm] | $\delta^{13}C$ VPDB ‰ | Collection method/site |
|---|---|---|---|
| 2021-11-21 | 435 | -7.46 | Ground |
| 2021-11-21 | 472 | -8.34 | Ground |
| 2021-11-21 | 437 | -7.65 | Ground |
| 2021-11-21 | 416 | -8.00 | Ground |
| 2021-11-28 | 671 | -4.44 | UAS |
| 2021-11-30 | 1030 | -3.65 | Ground |
| 2021-11-30 | 2998 | -2.12 | Ground |
| 2021-11-30 | 2863 | -2.15 | Ground |
| 2021-12-01 | 4459 | -2.03 | Ground |
| 2021-12-01 | 2722 | -1.47 | Ground |
| 2021-12-01 | 1326 | -2.40 | Ground |

velocities yielded results ranging from 10.0 to 12.2 $\mathrm{ms^{-1}}$ with an average of 11.1 $\mathrm{ms^{-1}}$. The wind direction given by the ERA5 model yielded results ranging from 38° to 68° with an average of 53°. These ranges contribute to the overall uncertainty $\epsilon_d$

### 3.2 Carbon isotopes of plume $CO_2$

The $CO_2$ concentrations and $\delta^{13}C$ values of plume gas samples are given in Table 2. Samples collected from the ground at the UNM Multi-GAS site show background $CO_2$ concentrations 416 to 471 ppm $CO_2$ with $\delta^{13}C$ values of -8‰ (relative to Peedee belemnite) which is close to that of air. The sample collected by UAS has a $CO_2$ concentration distinctly elevated from air of 671 ppm and a heavier $\delta^{13}C$ value of -4.44 ‰. Samples collected from the ground closer to the vent have even higher $CO_2$ concentrations from 1030 to 4459 ppm with $\delta^{13}C$ values from -2.40 to -1.47 ‰.

### 3.3 Multi-GAS measurements of plume

The Multi-GAS $CO_2/SO_2$ ratios during the period from November 21 to November 25, 2021 range from 5 to 26 and are shown in Table 2. These values are consistent with those reported by (Albertos et al., 2022) and (Burton et al., 2023). We use the range of reported $SO_2$ fluxes (mean of $10^4$ t day$^{-1}$ over the duration of the active eruption (Albertos et al., 2022)) in combination with the range of our Multi-GAS $CO_2/SO_2$ ratios to obtain $CO_2$ fluxes ranging from $7.3 \times 10^4$ to $3.6 \times 10^5$ t $CO_2$ day$^{-1}$ for 170 this period (Table 3).

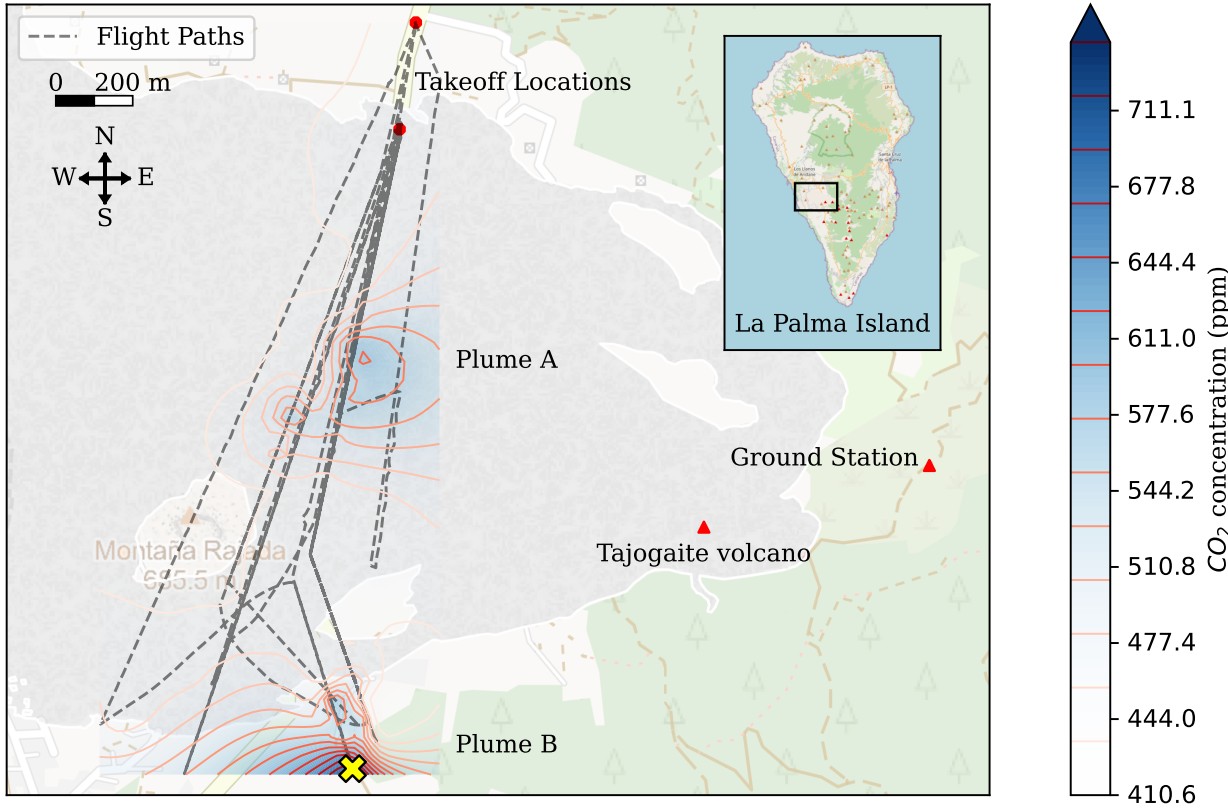

**Figure 2. Top-down perspective map of all transect flight paths.** Flights occurred over a four-day period during the 2021 eruption. This map includes a horizontal cross-section Kriging plot of the $CO_2$ concentration highlighted as the distinct Plumes A and B. The sample collection location is indicated by the yellow $\times$. Insert shows the location of Tajogaite Volcano on La Palma Island.

**Table 3.** Multi-GAS measurements, $SO_2$ flux and computed $CO_2$ flux .

| Date | Average $CO_2/SO_2$ (molar) | $SO_2$ flux (t/day) | $CO_2$ t/day |
|---|---|---|---|
| 2021-11-21 | $26 \pm 15$ | $2 \pm 1 \times 10^4$ | $3.6 \pm 1.8 \times 10^5$ |
| 2021-11-22 | $10 \pm 2$ | $2 \pm 1 \times 10^4$ | $1.4 \pm 0.7 \times 10^5$ |
| 2021-11-23 | $5 \pm 2$ | $2 \pm 1 \times 10^4$ | $7.3 \pm 3.7 \times 10^4$ |
| 2021-11-24 | $7 \pm 2$ | $2 \pm 1 \times 10^4$ | $9.5 \pm 4.8 \times 10^4$ |
| 2021-11-25 | $16 \pm 2$ | $2 \pm 1 \times 10^4$ | $2.3 \pm 1.1 \times 10^5$ |

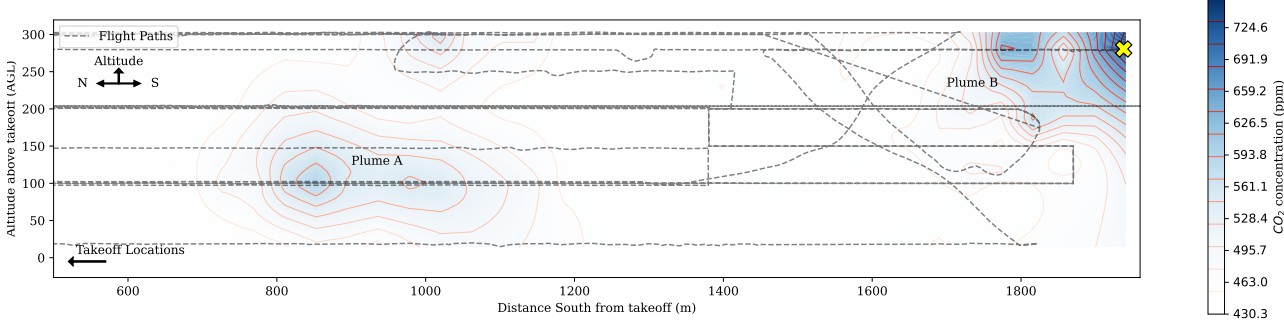

**Figure 3. Lateral perspective kriging map of all transects plotted in Figure 2.** The plot indicates two separate plumes in the vertical cross-section labeled Plume A and Plume B. The sample collection location is indicated by the yellow $\times$.

## 4 Discussion

This work highlights our efforts collecting and analysing $CO_2$ gasses during the Tajogaite volcanic eruption. Through this work, we demonstrated the efficacy of using a UAS to study the $CO_2$ plumes associated with an in-process eruption.

### 4.1 $CO_2$ Emissions

Our UAS-based $CO_2$ emission estimation technique yields $CO_2$ fluxes using direct measurement with a single type of instrument. This simplifies the estimation of $CO_2$ flux. However, in-situ measurement during an active eruption is challenging. The most serious difficulty we encountered was obtaining complete transects across the plume or plumes. In several of our transects, especially for the more distant Plume B, we were not successful in flying the UAS far enough to get to background $CO_2$ on the far side of the plume. Gas plumes change shape and direction on relatively short-time scales as the wind shifts. While ideally,

we would like to perform several flights at various altitudes through a plume in order to obtain a complete $CO_2$ concentration map of the plume, this is challenging for wide or distant plumes because of limited UAS flight times and the need to know the plume's location and extent a priori. To address this challenge we assume a Gaussian plume and fit a Gaussian curve to our data. We then rotate the Gaussian fit to obtain a 2D concentration slice which is multiplied with estimated wind velocity to yield the flux. This approach produces the most accurate results if we transect the plume through its widest part. However,

identifying the widest part and then transecting the plume before the plume changes will require teams of collaborating UASs. A good fit of the data by the Gaussian model is given by a high $R^2$ value. For instance, transect 2 was fit with a $R^2$ value of 0.93 accounts for 93% of the variance in the observed data. The model fit represented by this high $R^2$ value is depicted in Figure 4.

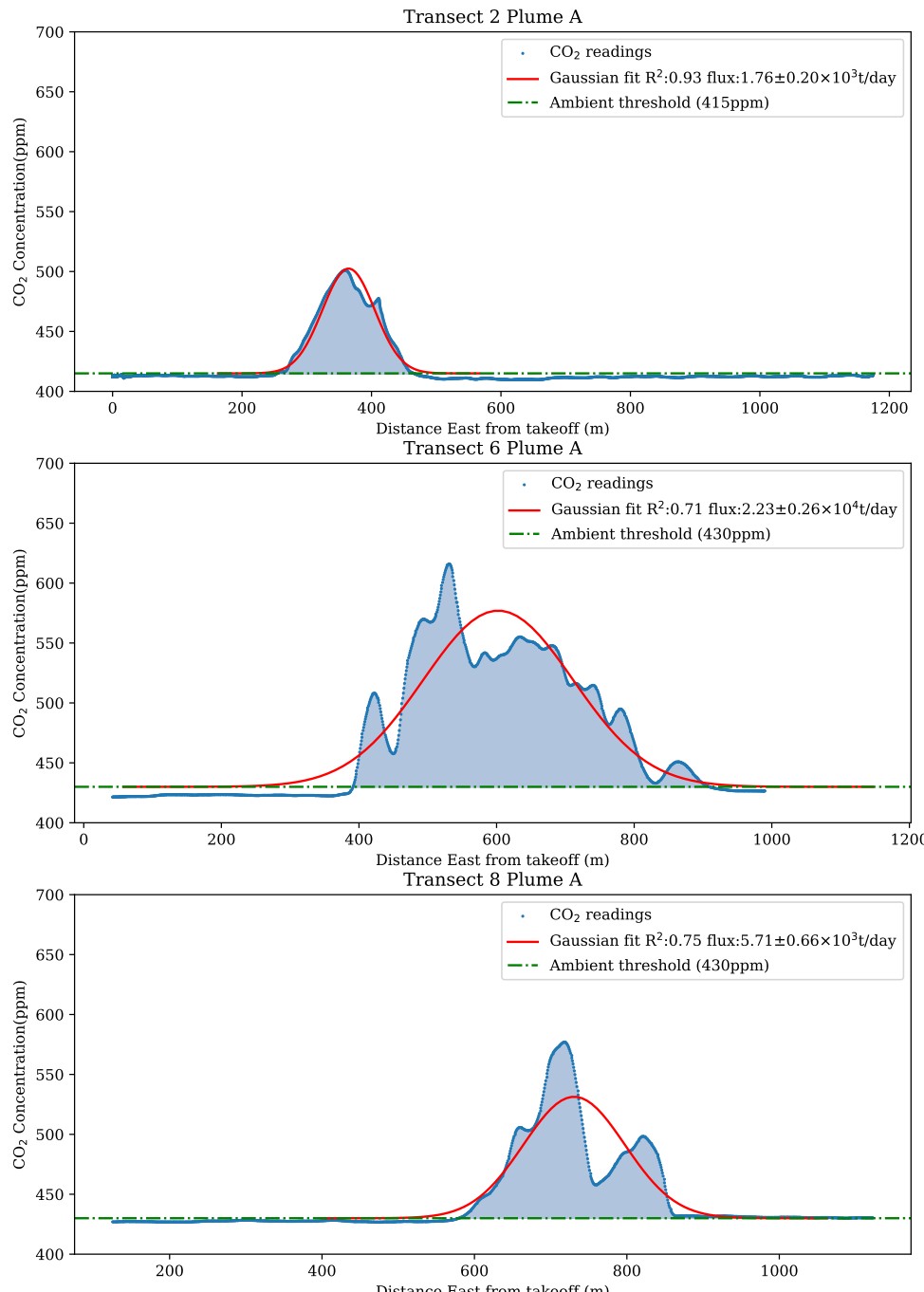

**Figure 4.** Three plots of encounters with plume A with the closest Gaussian model fit. $CO_2$ concentration (blue) over the encountered plume as a function of distance from takeoff location.

Uncertainty is introduced by the assumptions made by the model. With just one horizontal transect, we assume the vertical Gaussian standard deviation is identical to the horizontal standard deviation of the plume. Both dimension standard deviations are linearly correlated to the flux calculation, meaning that a 20% error in the vertical standard deviation will affect the flux estimate by 20%. We estimate the vertical standard deviation is likely close to the horizontal standard deviation, but the difference is impossible to determine. Additionally, we assume that the horizontal transect samples the plume at the altitude where the plume is widest. If the transect is not through the largest cross-section, the flux calculation may be a lower bound. Wind velocity was measured during one of the transects, but weather is notoriously unpredictable. This represents another source of uncertainty in the model which has a linear effect on the flux measurement. We used our wind estimates during the time of each flux calculation. This variation in wind velocity $\epsilon_v$ is $\pm 11\%$ which is calculated from the wind velocity range measured over the experiments (Table 1). The range of wind directions is $\pm 15°$ from Table 1, which gives an error in the flux estimate based on $\epsilon_d = 1 - \cos(\text{angle})$, thus $\pm 3.40\%$. The SBA-5 documentation reports sensor error $\epsilon_s$ is $1\%$ in the range of $CO_2$ we measured. Finally, background ambient $CO_2$ $\epsilon_b$ adds $1\%$ to the uncertainty model which we calculated from the uncertainty in ambient $CO_2$ readings. Therefore, our estimated flux uncertainty given by the root sum of squares method is $\epsilon = \pm 11.61\%$.

Our data show that for Plume A, transect 6 (Figure 3) represents the widest plume and results in the highest $CO_2$ flux value of $2.23 \pm 0.26 \times 10^4 \, \text{t day}^{-1}$, an order of magnitude higher than the other two Plume A transects. This transect was flown at the lowest altitude (100 m) of the three, implying that the other two transects only captured the upper parts of the plume. Comparison with $CO_2$ fluxes obtained by combining $SO_2$ fluxes with $CO_2$ to $SO_2$ ratios measured 1 km from the vent gives fluxes ranging from $7.3 \times 10^4$ to $3.6 \times 10^5 \, \text{t} \, CO_2 \, \text{day}^{-1}$ (Table 3). Therefore our highest flux measurement is consistent with the lowest estimate using the combined method. While comparing these two approaches is helpful, our experiment was not designed to make a direct comparison. The discrepancy could be due to a significantly varying $CO_2$ emission rate during eruptions, an overestimate of the $SO_2$ flux, or the lack of validity of the 2D Gaussian extrapolation approach. Our estimates are consistent with the October 2021 high emissions presented by Bruton et al., 2023 who report fluxes of $3.36 \times 10^4$ to $4.19 \times 10^4 \, \text{t} \, CO_2 \, \text{day}^{-1}$ (389 to 486 kg/s) for the smaller, non-ashy plume that we measured. More work needs to be performed in the future to better assess sources of discrepancies with new and coordinated measurements at passively degassing and erupting volcanoes. However, even with such discrepancies, it is clear that the Tajogaite eruption in November 2021 produced a $CO_2$ flux up to $2 \times 10^4 \, \text{t day}^{-1}$ or even $5 \times 10^5 \, \text{t day}^{-1}$. Even the $5 \times 10^5 \, \text{t day}^{-1}$ would be only 0.4% of the daily $CO_2$ emitted by the burning of fossil fuels (Conlen, 2021).

## 4.2 Carbon Isotopes

The carbon isotope data obtained from the UAS-captured samples and the samples collected from the ground are generally consistent and show mixing of air-derived $CO_2$ with a deep magmatic source. Figure 5 shows that all plume samples collected from the ground define a set of mixing lines in $\delta^{13}C$ versus $CO_2^{-1}$ space, i.e. in a Keeling plot (Keeling, 1958) that allows for the extrapolation of the $\delta^{13}C$ value of the pure $CO_2$ being emitted from the volcanic vent. The sample collected by UAV lies slightly above this set of mixing lines and extrapolates to somewhat heavier $\delta^{13}C$. The resulting volcanic $\delta^{13}C$ values

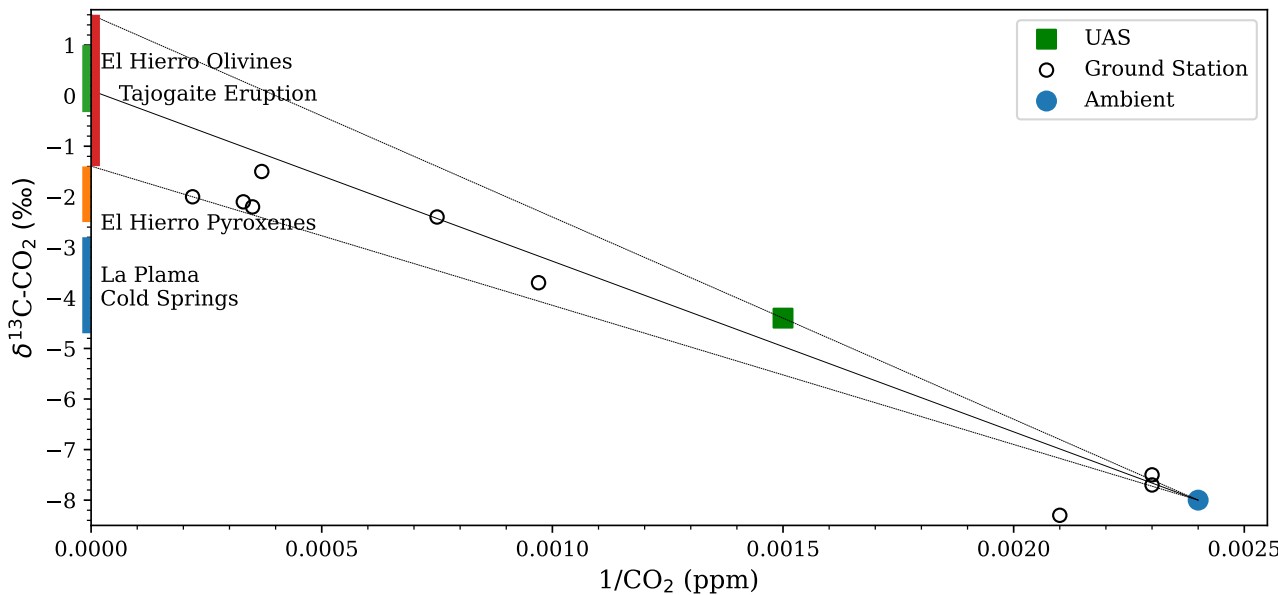

**Figure 5. Keeling plot showing standard air, samples collected on the ground, and with the UAS.** Linear extrapolation indicates a volcanic $\delta^{13}C - CO_2$ value of -1.40 to 1.60 ‰. Also shown are data from olivines and pyroxenes collected at the El Hierro Volcano (Sandoval-Velasquez et al., 2021) and the composition of cold $CO_2$-rich gas discharges on La Palma Island (Padrón et al., 2015).

taking into account all samples lies between -1.5 and +1.5 ‰. Despite these uncertainties, these values overlap with $\delta^{13}C$ data obtained from mantle xenoliths erupted at the nearby El Hierro Volcano (Sandoval-Velasquez et al., 2021). Extrapolation of all

225   these data results in a $\delta^{13}C$ value of $0.1\pm1.5$‰. Notably the carbon isotope values are significantly heavier than those measured in cold $CO_2$-rich gas discharges from springs on La Palma (Padrón et al., 2015) and within the range of values measured in olivines and pyroxenes of xenoliths from El Hierro Island (Sandoval-Velasquez et al., 2021). These authors suggested that the heavy values of the xenoliths are related to recycling of crustal carbon, likely derived from carbonates into the mantle source of the Canary Islands hot spot. Our data suggests that the magmatic system that is driving the Tajogaite eruption taps into this

230   deep $CO_2$, rather than remobilizing $CO_2$ that feeds the cold degassing springs on the island. Sandoval-Velasquez et al. (2024) report $\delta^{13}C$ values measured in olivines, clinopyroxenes and orthopyroxenes from lava flows erupted in 2021. Their data is consistent with our extrapolated heavy $\delta^{13}C$ values. For olivines, representing the earliest crystallization phase, their values range from 0 to 1‰. Values are somewhat lighter for orthopyroxenes and clinopyroxenes. Using all data, their estimated mantle endmember is -1.5‰. Our data extrapolate to -1.4 to +1.6‰. Given the difference in sample medium, i.e. phenocrysts versus

235   gas plume, the results are remarkably consistent. More work at erupting volcanoes is needed to better constrain the sources of magmatic $CO_2$ emitted during heightened activity of volcanic systems.

## 5 Conclusion

The use of UAS is revolutionizing volcano science by enabling the collection of data that previously required extensive, costly, and hazardous aerial surveys using piloted fixed-wing aircraft or helicopters. Especially in the field of volcanic gases, recent UAS-based campaigns showed the value of utilizing UAS to make gas flux and gas composition measurements and also collect plume samples for subsequent chemical and isotopic analyses (Liu et al., 2020; Galle et al., 2021). Our work during the explosive and hazardous eruption of the Tajogaite Volcano shows that $CO_2$ emission measurements and plume gas samples can be collected even during these heightened periods of volcanic activity. We demonstrate that a UAS capable of automated sampling can be guided by the expert knowledge of scientists in the field to collect valuable data that would be impossible with robots or scientists alone. The collected data provide key insights into the volcano's state and the course of an eruption. Future work is needed to increase UAS autonomy in choosing flight paths to more completely capture data from dynamic plumes, but, as we have demonstrated, the present approach works for volcano monitoring during eruptions and can provide much-needed information about eruptive gas emissions.

*Code and data availability.* Additional data and plot generation code is available at https://github.com/BCLab-UNM/lapalma-expedition/tree/2021_tajogaite_eruption. UAS code available at https://github.com/BCLab-UNM/dragonfly-dashboard https://github.com/BCLab-UNM/dragonfly-controller.

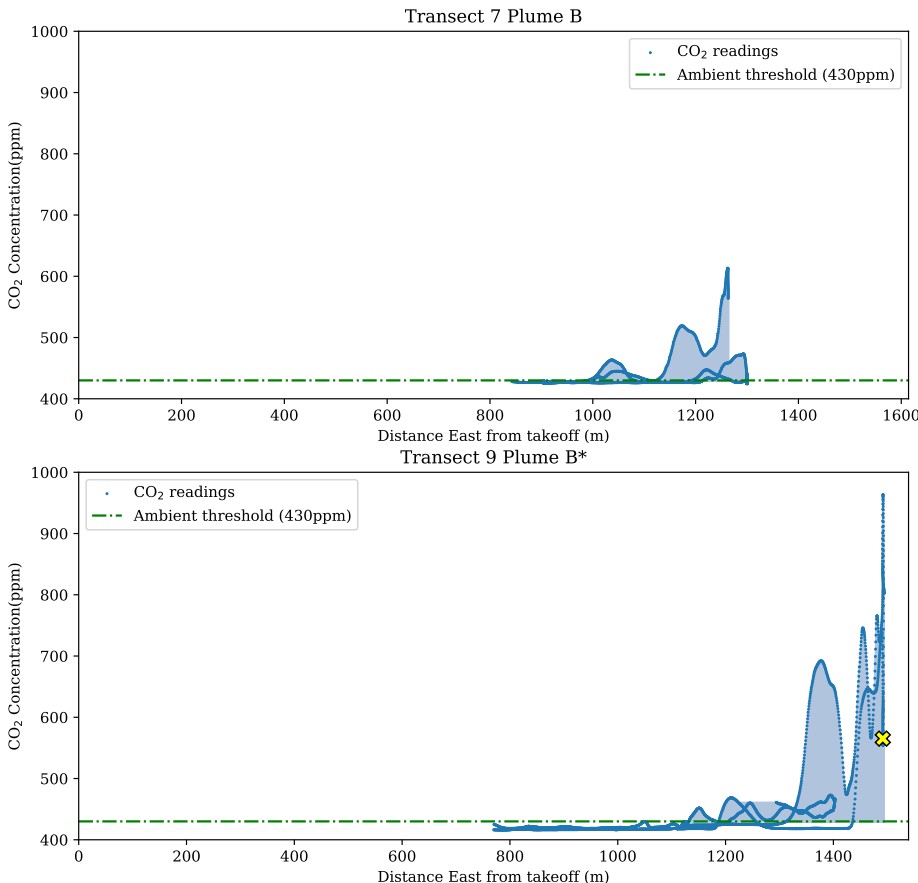

**Figure A1.** Encounters with plume B were not as well-fit as plume A encounters. These plots show the $CO_2$ readings collected during the two highest plume model fit. As with Figure 4, $CO_2$ concentration (blue) over the encountered plume as a function of distance from takeoff location. The sample collection location is indicated by the yellow ×.

*Author contributions.* Author contributions: JE, GMF, SN, and TF (UNM VolCAN team) performed UAS fieldwork for this paper. JE, NP, PHP, EPG (INVOLCAN team) and TF conducted the ground fieldwork. JE developed UAS software and hardware supervised by GMF and MEM. SN designed the sample collection device. JE, NP, PHP, EPG performed data analysis. TF performed isotope and gas analysis. JE, TF, GMF, SN, and MEM wrote the manuscript.

*Competing interests.* The authors declare that they have no competing interests. The authors give consent for publication. All data needed to evaluate the conclusions in the paper are present in the paper and/or the Supplementary Materials.

*Acknowledgements.* JE support provided by the Department of Energy's Kansas City National Security Campus, operated by Honeywell Federal Manufacturing & Technologies, LLC under contract number DE-NA0002839. GMF, SN, TF, RF, and MM support provided by the VolCAN project under National Science Foundation grant 2024520. Support was also provided by a Google CSR award. This study received funding from Google and Honeywell Federal Manufacturing & Technologies, LLC. VOLRISKMAC II (MAC2/3.5b/328) financed by the Program INTERREG V A Spain-Portugal MAC 2014-2020 of the European Commission. Thanks to Samantha Wolf for help calculating the flux.

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
