# Peer review of "Drone CO2 Measurements During the Tajogaite Volcanic Eruption"

_Atmospheric Measurement Techniques, 2023_

## Referee Comment (RC1)

**Review of "Drone $CO_2$ Measurements During the Tajogaite Volcanic Eruption" by Ericksen et al. (2024)**

The manuscript by Ericksen et al. (2024) applies Unpiloted Aerial System (UAS) platforms to measure carbon dioxide ($CO_2$) concentrations and carbon isotope ratios during the 2021 eruption of the Tajogaite Volcano in Spain. This study used a Dragonfly UAS outfit with systems for measuring $CO_2$ concentrations and carbon isotopic ratios for 10 transects through volcanic plumes during the eruption. Using measured $CO_2$ concentrations and winds, applying gaussian assumptions, led to emission rate estimates of $1.19 \times 10^6$ to $2.80 \times 10^7$ t day$^{-1}$ (1190 to 28000 kt day$^{-1}$). These are very large emission rates compared to recent literature estimates and 1-2 orders of magnitude larger compared to those derived using ground-based measurements in this study ($1.4 \times 10^4$ to $3.6 \times 10^5$ t day$^{-1}$). The study conducts no attempt to derive uncertainty/errors of these estimates or compare their results with past literature. The lack of description of the methods used in the study made it challenging to understand where errors could be coming from. The paper was very short so there is plenty of space to provide significantly more detail about the methods and results applied in this study. My suggestions for this are below. The paper has issues with grammar and typos and overall reads more as a report and less like a manuscript. The one results of the study I agree with is the fact that UAS systems are vital for measurements of trace gases in volcanic plumes, but this has been shown before (e.g., Xi et al., 2016). As is, this manuscript is not sufficient for publication in *Atmospheric Measurement Techniques*.

**Major Comments**

1. Line 36-37. I think the authors need to discuss more of the challenges with using satellite XCO2 to monitor volcanic plumes and estimate $CO_2$ fluxes. Satellites provide improved spatiotemporal observational coverage compared to ground-based and aircraft measurements for volcanic plumes. However, satellite XCO2 retrievals are associated with error due to aerosol and water vapor interference, measuring cumulative fluxes instead of direct volcanic emissions, and overall retrieval uncertainties. The work by Johnson et al. (2020) describes these issues in detail.

2. Line 92. These $CO_2/SO_2$ ratios are at the high end of reported values in the literature. The authors should compare their measurements to other studies in the literature and discuss this comparison in the paper.

3. Carbon isotope data. The authors need to describe what this data is and what it is used for to better understand volcanic sources of $CO_2$. Readers in atmospheric science, such as this reviewer, and other fields outside of volcanology will be interested in this manuscript; therefore, the authors need to better describe some of the data/terms used in this study.

4. Methods Section. This section needs to come before the results and discussion of the study. It is impossible to follow the results of the work if the reader has no idea about the tools, methods, uncertainties, etc. associated with the results.

5. Line 121. The study derives a maximum $CO_2$ flux of 4730 kt day$^{-1}$. This value seems pretty large. Can the authors compare this value to other studies from other volcanoes? Some context for all the results in this study is lacking. Upon reviewing this manuscript further, it appears the emission estimate results presented in the abstract ($1.19 \times 10^6$ to $2.80 \times 10^7$ t day$^{-1}$ (1190 to 28000 kt day$^{-1}$)) don't match what is presented in the body of the text ($1.65 \times 10^4$ to $4.73 \times 10^6$ t day$^{-1}$). I don't understand the discrepancy between the presented flux estimates. The manuscript discusses the results in such little detail it is challenging to follow.

6. Line 131-134. There are other studies that have applied UAS platforms to measure $SO_2$ and $CO_2$ in volcanic plumes (e.g., Xi et al., 2016) that are not discussed in this manuscript. It would be useful to expand upon the importance of UAS platforms for monitoring volcanic plumes and compare the results of this study to past work. Overall, there appears to be a lack of literature review and comparison of the author's findings with past work discussed throughout this manuscript.

7. Line 150-151. What does this sentence mean? How does the UAS drift with the plume? Is this done physically or using near-real-time concentration measurements to remain within the regions of maximum $CO_2$ concentrations? Much more information and details are needed about the UAS capabilities and how it was used in this study.

8. Line 166. What is the ambient/background $CO_2$ concentration derived in this study? Also, how was it derived? This value is critical for estimating $\Delta CO2$ concentrations and the corresponding emission rates. Also, what are the uncertainty levels associated with these ambient/background $CO_2$ concentration estimates?

9. Line 167. Is the gaussian assumption for the volcanic plume shape appropriate here for these flux estimates during all 10 transects? The authors should plot the $CO_2$ concentrations throughout the transect of the volcanic plumes monitored here. This could easily show whether or not the plumes measured in this case were in fact close to gaussian in shape. If not, the gaussian assumption might not be appropriate for these flux estimates.

After reading the rest of the paper I found Figure A1. Why was this figure not referenced in the text of the manuscript? This figure is probably more important than figures such as Fig. 4. Maybe move this figure to the main body of the text or make sure to reference it clearly.

That being said, transect 2 and maybe 6 measure volcanic plumes that are close to gaussian in shape. However, some of the other transects clearly show non-gaussian characteristics. How does this impact the estimated emission rates presented in this study?

10. Line 175. What are the uncertainty levels associated with the measured wind speeds and direction? Errors associated with measured winds can be large. Also, how variable were winds throughout the plume? How was this variability treated in the flux calculations? This was not discussed at all and is needed in order to reproduce and assess the results of this manuscript.

11. Uncertainty. This study lacks any discussion, or attempt to determine, uncertainty levels associated with the flux results derived in this study. The emission rates are quite high in some of the transects and these results need to be assessed for error/uncertainty and more thoroughly compared to past studies. Errors/uncertainty from wind speed measurements (wind speeds aren't displayed or discussed in the study), gaussian assumptions, plume extrapolation methods, $CO_2$ concentration measurements (sensor uncertainty and the lack of observational coverage of the sensor in each transect), and other potential error sources can be large. These values need to be quantified in order to understand these results.

12. Derived emission rates. This study estimated very large $CO_2$ emission rate estimates (those provided in the abstract) of $1.19 \times 10^6$ to $2.80 \times 10^7$ t day$^{-1}$ (1190 to 28000 kt day$^{-1}$). These estimates are 1-2 orders of magnitude larger compared to those derived from ground-based estimates in this study ($1.4 \times 10^4$ to $3.6 \times 10^5$ t day$^{-1}$). The authors state that these differences could be due to emission variability, underestimate of the $SO_2$ flux, or the lack of validity of the 2D gaussian assumptions. The authors need to look into these discrepancies much closer in order to trust the results of the $CO_2$ emissions stated in this study. Also, these derived values are much larger than other estimates in the literature from other volcanoes. The authors need to carefully investigate the literature to see if they can find $CO_2$ emission rates at the magnitude of those derived in this study.

**Minor Comments**

1. Line 9-10. A forecasting signal for what? An impending eruption? This sentence seems incomplete.

2. Line 11. Remove "gas" from the beginning of this sentence.

3. Line 11. Instead of "steam" I think you mean "water vapor".

4. Line 17. This is the first time "UAS" has been used in the body of the text, therefore you should define the abbreviation here.

5. Line 67-68. Are the authors referring to the TROPOMI sensor data onboard the Sentinel 5 Precursor satellite?

6. Line 66-79. It seems like it would be easier for the readers to follow the emission rate estimate discussions if the authors used consistent units. Can the authors just use kt day$^{-1}$ for both $SO_2$ and $CO_2$ instead of using $\times 10^6$ or $\times 10^4$ t day$^{-1}$?

7. Line 79-80. What does this hover-drift test wind speed value mean? Is this an average wind speed measured during the 10 flights? This needs some further explanation.

8. Line 79. Do you mean Fig. 1? Same thing when referencing Table 1 for the first time in Line 81.

9. Line 82. Just use the actual $CO_2$ values in ppm and not the scientific notation of the values.

10. Line 86. Figure 5 should be Figure 2. The authors should reference tables and figures in sequential order.

11. Line 93. I think some values are missing in this sentence.

12. Line 144. "To" instead of "TO".

13. Line 150. Is "(2)" trying to make a reference to Figure 2?

**References**

Johnson, M. S., Schwandner, F. M., Potter, C. S., Nguyen, H. M., Bell, E., Nelson, R. R., et al. (2020). Carbon dioxide emissions during the 2018 Kilauea volcano eruption estimated using OCO-2 satellite retrievals. Geophysical Research Letters, 47, e2020GL090507. https://doi.org/10.1029/2020GL090507.

Xi, X., Johnson, M. S., Jeong, S., Fladeland, M., Pieri, D., Diaz, J. A., et al. (2016). Constraining the sulfur dioxide degassing flux from Turrialba volcano, Costa Rica using unmanned aerial system measurements. Journal of Volcanology and Geothermal Research, 325, 110–118. https://doi.org/10.1016/j.jvolgeores.2016.06.

---

## Author Response (AR1)

**Review of "Drone CO₂ Measurements During the Tajogaite Volcanic Eruption" by Ericksen et al. (2024)**

**Reviewer Summary:** The manuscript by Ericksen et al. (2024) applies Unpiloted Aerial System (UAS) platforms to measure carbon dioxide ($CO_2$) concentrations and carbon isotope ratios during the 2021 eruption of the Tajogaite Volcano in Spain. This study used a Dragonfly UAS outfit with systems for measuring $CO_2$ concentrations and carbon isotopic ratios for 10 transects through volcanic plumes during the eruption. Using measured $CO_2$ concentrations and winds, applying gaussian assumptions, led to emission rate estimates of $1.19 \times 10^6$ to $2.80 \times 10^7$ t day$^{-1}$ (1190 to 28000 kt day$^{-1}$). These are very large emission rates compared to recent literature estimates and 1-2 orders of magnitude larger compared to those derived using ground-based measurements in this study ($1.4 \times 10^4$ to $3.6 \times 10^5$ t day$^{-1}$). The study conducts no attempt to derive uncertainty/errors of these estimates or compare their results with past literature. The lack of description of the methods used in the study made it challenging to understand where errors could be coming from. The paper was very short so there is plenty of space to provide significantly more detail about the methods and results applied in this study. My suggestions for this are below. The paper has issues with grammar and typos and overall reads more as a report and less like a manuscript. The one result of the study I agree with is the fact that UAS systems are vital for measurements of trace gases in volcanic plumes, but this has been shown before (e.g., Xi et al., 2016). As is, this manuscript is not sufficient for publication in *Atmospheric Measurement Techniques*.

**Author Response:**

Thank you for the detailed review. We found the comments very helpful and have adapted them into our submission. In particular, we have 1) reevaluated our method for estimating $CO_2$ flux and have a new value that is in line with the $SO_2/CO_2$ ratio estimate, 2) we have included more detailed information on inputs to our model, such as the source of the wind speed estimates and ambient $CO_2$ concentrations, 3) we have included a more detailed related works section, 4) more detailed methods regarding the UAS and data collection, and 5) we have performed a thorough edit to remove typos.

**Major Comments**

1. Line 36-37. I think the authors need to discuss more of the challenges with using satellite XCO2 to monitor volcanic plumes and estimate $CO_2$ fluxes. Satellites provide improved spatiotemporal observational coverage compared to ground-based and aircraft measurements for volcanic plumes. However, satellite XCO2 retrievals are associated with error due to aerosol and water vapor interference, measuring cumulative fluxes instead of direct volcanic emissions, and overall retrieval uncertainties. The work by Johnson et al. (2020) describes these issues in detail.

**Response:**

We address these points in more detail in the revised manuscript. We include the text: "The study by Johnson et al., 2020 utilized the Orbiting Carbon Observatory -2 (OCO-2) to measure the $CO_2$ emissions from the 2018 Klīlauea eruption. Their measurements of 77.1±41.6 kt/day were made during one day of observations where conditions were ideal to enable consistent high-quality data during the traverse of the majority of the plume. Cloud coverage and aerosol are the major inhibitors for obtaining consistent $CO_2$ data using OCO-2. In addition, the wind direction must be near perpendicular to the satellite's orbit path and the measurements must be made down-wind from the plume. The OCO-2 16-day repeat cycle currently makes this method impractical for frequent, high-rate $CO_2$ flux measurements from erupting volcanoes and the only other successful volcanic $CO_2$ emission study is that of Schwandner et al., 2017 from Yasur in Vanuatu."

2. Line 92. These $CO_2/SO_2$ ratios are at the high end of reported values in the literature. The authors should compare their measurements to other studies in the literature and discuss this comparison in the paper.

**Response:**

We report the $CO_2/SO_2$ data previously published by Albertos et al., 2022. The reported ratios have a range of 5-22 molar. We include our own measurements in the revised version in a Table . Our average $CO_2/SO_2$ molar ratios measured during Nov. 21 to 25, 2021 range from 5 to 26. These data are entirely consistent with the recently reported $CO_2/SO_2$ data measured by multiGAS and also by FTIR and reported by Burton et al., 2023. Their data for this time period ranges from 2 – 40. With most measurements from 10 – 40 over their entire observation period (mid-September to late November, 2021). Therefore, our data is consistent with other measurements made at Tajogatite Volcano.

3. Carbon isotope data. The authors need to describe what this data is and what it is used for to better understand volcanic sources of $CO_2$. Readers in atmospheric science, such as this reviewer, and other fields outside of volcanology will be interested in this manuscript; therefore, the authors need to better describe some of the data/terms used in this study.

**Response:**

We have expanded this section to now include more information about this data and what it is used for. We also cite recent work by Sandoval-Velasques et al., 2024. These workers report the $\delta^{13}C$ values of olivine, CPX and OPX in lava flows from the 2021 eruption. Their data is consistent with our extrapolated heavy $\delta^{13}C$ values. For olivines, representing the earliest crystallization phase, values range from 0 to 1‰. Values are somewhat lighter for OPX and CPX. Using all data, their estimated mantle endmember is -1.5‰. Our data extrapolate to -1.4

to + 1.6 ‰. Given the difference in sampling and sample medium, i.e. phenocrysts versus gas plume, the results are remarkably consistent.

Sandoval-Velasquez, A., Casetta, F., Ntaflos, T., Aiuppa, A., Coltorti, M., Frezzotti, M.L., Alonso, M., Padrón, E., Pankhurst, M., Pérez, N.M. and Rizzo, A.L., 2024. 2021 Tajogaite eruption records infiltration of crustal fluids within the upper mantle beneath La Palma, Canary Islands. Frontiers in Earth Science, 12.

4. Methods Section. This section needs to come before the results and discussion of the study. It is impossible to follow the results of the work if the reader has no idea about the tools, methods, uncertainties, etc. associated with the results.

**Response:**

We have reorganized the paper so that the Methods are presented before the Results section.

5. Line 121. The study derives a maximum $CO_2$ flux of 4730 kt day$^{-1}$. This value seems pretty large. Can the authors compare this value to other studies from other volcanoes? Some context for all the results in this study is lacking. Upon reviewing this manuscript further, it appears the emission estimate results presented in the abstract ($1.19 \times 10^6$ to $2.80 \times 10^7$ t day$^{-1}$ (1190 to 28000 kt day$^{-1}$)) don't match what is presented in the body of the text ($1.65 \times 10^4$ to $4.73 \times 10^6$ t day$^{-1}$). I don't understand the discrepancy between the presented flux estimates. The manuscript discusses the results in such little detail it is challenging to follow.

**Response:**

We have revised the methods of our $CO_2$ flux calculation and extended the methods section to reflect this. The method is now explained in more detail and the revised $CO_2$ fluxes are $3.88 \pm 0.38 \times 10^3$ (95%CI) t/day for plume A and $2.85 \pm 0.23 \times 10^4$ (95%CI) t/day for plume B. These are consistent with the recent high fluxes presented by Bruton et al., 2023 who report fluxes of $3.36 \times 10^4$ - $4.19 \times 10^4$ t/day (389 – 486 kg/s) for the smaller, non-ashy plume that we also measured during our work. We have included the comparison to the Burton et al. results in the discussion section.

6. Line 131-134. There are other studies that have applied UAS platforms to measure $SO_2$ and $CO_2$ in volcanic plumes (e.g., Xi et al., 2016) that are not discussed in this manuscript. It would be useful to expand upon the importance of UAS platforms for monitoring volcanic plumes and compare the results of this study to past work. Overall, there appears to be a lack of literature review and comparison of the author's findings with past work discussed throughout this manuscript.

**Response:**

We are well aware of the Xi et al., 2016 paper. This paper focuses on measuring $SO_2$ emissions

using a fixed-wing UAS and a tethered balloon. They did use electrochemical sensors but only for $SO_2$ concentration measurements and not for $CO_2$. We feel that we have adequately discussed this paper but have included other recent papers that report UAS-based methods in the related work section:

Rüdiger, J., Tirpitz, J.L., de Moor, J.M., Bobrowski, N., Gutmann, A., Liuzzo, M., Ibarra, M. and Hoffmann, T., 2018. Implementation of electrochemical, optical and denuder-based sensors and sampling techniques on UAV for volcanic gas measurements: examples from Masaya, Turrialba and Stromboli volcanoes. Atmos. Meas. Tech., 11(4): 2441-2457.

These authors use mulitGAS, denuder sampler and miniDOAS systems.

Stix, J., de Moor, J.M., Rüdiger, J., Alan, A., Corrales, E., D'Arcy, F., Diaz, J.A. and Liotta, M., 2018. Using Drones and Miniaturized Instrumentation to Study Degassing at Turrialba and Masaya Volcanoes, Central America. Journal of Geophysical Research: Solid Earth, 123(8): 6501-6520.

These authors used multiGAS and miniDOAS systems.

7. Line 150-151. What does this sentence mean? How does the UAS drift with the plume? Is this done physically or using near-real-time concentration measurements to remain within the regions of maximum $CO_2$ concentrations? Much more information and details are needed about the UAS capabilities and how it was used in this study.

**Response:**

The Dragonfly was used to measure the plume velocity by drifting in the plume, i.e. being carried down-wind by the wind that transports the plume. This is a common method used to estimate plume velocities. (Liu et al., 2020; Galle et al., 2021).

We also measured the wind with a hand-held manometer during the same time we used the drift method. The manometer measurements returned a wind speed of 10 - 12 m/sec, entirely consistent with our drift method. We included have included these details in our revised version.

8. Line 166. What is the ambient/background $CO_2$ concentration derived in this study? Also, how was it derived? This value is critical for estimating $\Delta CO_2$ concentrations and the corresponding emission rates. Also, what are the uncertainty levels associated with these ambient/background $CO_2$ concentration estimates?

**Response:**

We have included ambient background $CO_2$ measurements. These are shown in Figure A1. We set the ambient background $CO_2$ to 430 ppm in our computations. The actual measurements of

ambient $CO_2$ well outside of the plume (up to 400 m away from the edge of the plume) are very consistent and vary only from about 425 – 430 ppm. The plume $CO_2$ content is clearly above this background.

9. Line 167. Is the gaussian assumption for the volcanic plume shape appropriate here for these flux estimates during all 10 transects? The authors should plot the $CO_2$ concentrations throughout the transect of the volcanic plumes monitored here. This could easily show whether or not the plumes measured in this case were in fact close to gaussian in shape. If not, the gaussian assumption might not be appropriate for these flux estimates.

After reading the rest of the paper I found Figure A1. Why was this figure not referenced in the text of the manuscript? This figure is probably more important than figures such as Fig. 4. Maybe move this figure to the main body of the text or make sure to reference it clearly.

That being said, transect 2 and maybe 6 measure volcanic plumes that are close to gaussian in shape. However, some of the other transects clearly show non-gaussian characteristics. How does this impact the estimated emission rates presented in this study?

**Response:**

We agree and have put Figure A1 in the main text. We have also included a discussion of the impact of non-Gaussian characteristics on the emission rates, we refer to this figure in this discussion. The goodness of the fit is qualitatively evaluated using the chi-square value. A good fit has a lower chi-square than a poor fit. However, quantifying the goodness of fit into a quantitative error in the flux value is not possible without detailed mapping of the entire cross-section of the plume which is beyond the scope of our study.

10. Line 175. What are the uncertainty levels associated with the measured wind speeds and direction? Errors associated with measured winds can be large. Also, how variable were winds throughout the plume? How was this variability treated in the flux calculations? This was not discussed at all and is needed in order to reproduce and assess the results of this manuscript.

**Response:**

Thank you for this comment. We were not able to measure wind speed throughout the plume but were able to measure it using two independent methods: 1) the UAV drift method that is described under point 7 of this reply and 2) a method that uses a hand-held manometer. The results from both methods are consistent. The drift method returned a wind-speed of 10.7 m/s and the manometer-based method resulted in measurements ranging from 10 - 12 m/s with an average of 11 m/s. As the wind-speed scales linearly at a 1:1 factor with the plume $CO_2$ concentration when calculating flux, the uncertainty associated with the wind-speed estimates is ± 10%. We have included this in the revised text.

11. Uncertainty. This study lacks any discussion, or attempt to determine, uncertainty levels

associated with the flux results derived in this study. The emission rates are quite high in some of the transects and these results need to be assessed for error/uncertainty and more thoroughly compared to past studies. Errors/uncertainty from wind speed measurements (wind speeds aren't displayed or discussed in the study), gaussian assumptions, plume extrapolation methods, $CO_2$ concentration measurements (sensor uncertainty and the lack of observational coverage of the sensor in each transect), and other potential error sources can be large. These values need to be quantified in order to understand these results.

**Response:**

We added error ranges and supporting information for the reason for the ranges to the discussion. You can find the updated text in the $CO_2$ Emissions subsection, on page 11. There are two sources of error. The first is error from uncertainties in the model, and the second is uncertainties from the data. We found uncertainties in the model to be linear in the shape of the Gaussian plume, both the horizontal standard deviation and the vertical standard deviation. The uncertainty from the data is dominated by the measured wind speed and this has a linear effect on the final flux calculation. From wind speed measurements during the period, we found the wind to deviate by as much as 1 m/s, between 10 m/s and 12 m/s. This gives us an error range of $\pm$ 10% on the final flux calculation. The uncertainties in the $CO_2$ measurements and the extrapolations are discussed in the response to point 12.

12. Derived emission rates. This study estimated very large $CO_2$ emission rate estimates (those provided in the abstract) of $1.19 \times 10^6$ to $2.80 \times 10^7$ t day$^{-1}$(1190 to 28000 kt day$^{-1}$). These estimates are 1-2 orders of magnitude larger compared to those derived from ground-based estimates in this study ($1.4 \times 10^4$ to $3.6 \times 10^5$t day$^{-1}$). The authors state that these differences could be due to emission variability, underestimate of the $SO_2$ flux, or the lack of validity of the 2D gaussian assumptions. The authors need to look into these discrepancies much closer in order to trust the results of the $CO_2$ emissions stated in this study. Also, these derived values are much larger than other estimates in the literature from other volcanoes. The authors need to carefully investigate the literature to see if they can find $CO_2$ emission rates at the magnitude of those derived in this study.

**Response:**
We thank the reviewer for catching this error. Upon further inspection we discovered a mathematical error in our calculations, resulting in a much lower flux value. We fixed this error, updated the methods section with this fix, and updated the resulting flux values, tables, and figures. Our updated flux values are $3.89 \pm 0.38$ x$10^3$ to $2.33 \pm 0.23$ x $10^4$.

**Minor Comments**

1. Line 9-10. A forecasting signal for what? An impending eruption? This sentence seems

incomplete.

**Response:**

We have modified the sentence and highlighted the importance of these measurements to understand magma dynamics.

2. Line 11. Remove "gas" from the beginning of this sentence.

**Response:**

Fixed.

3. Line 11. Instead of "steam" I think you mean "water vapor".

**Response:**

Fixed.

4. Line 17. This is the first time "UAS" has been used in the body of the text, therefore you should define the abbreviation here.

**Response:**

Fixed.

5. Line 67-68. Are the authors referring to the TROPOMI sensor data onboard the Sentinel 5 Precursor satellite?

**Response:**

Yes the Sentinel 5P TROPOMI data is referenced. It has been edited to specify TROPOMI and Sentinel 5P.

6. Line 66-79. It seems like it would be easier for the readers to follow the emission rate estimate discussions if the authors used consistent units. Can the authors just use kt day$^{-1}$ for both $SO_2$ and $CO_2$ instead of using $\times 10^6$ or $\times 10^4$ t day$^{-1}$?

**Response:**

Fixed - we now use t day$^{-1}$ for everything for consistency

7. Line 79-80. What does this hover-drift test wind speed value mean? Is this an average wind speed measured during the 10 flights? This needs some further explanation.

**Response:**

We have responded to this in point 7.

8. Line 79. Do you mean Fig. 1? Same thing when referencing Table 1 for the first time in Line 81.

**Response:**

We fixed this, it was referencing Table 1.

9. Line 82. Just use the actual $CO_2$ values in ppm and not the scientific notation of the values.

**Response:**
Agreed and fixed.

10. Line 86. Figure 5 should be Figure 2. The authors should reference tables and figures in sequential order.

**Response:**

Fixed

11. Line 93. I think some values are missing in this sentence.

**Response:**

Fixed

12. Line 144. "To" instead of "TO".

**Response:**

Fixed.

13. Line 150. Is "(2)" trying to make a reference to Figure 2?

**Response:**

Fixed, it was referencing Figure 2.

**Review of the article "Drone CO2 measurements during the Tajogaite Volcanic eruption"**

**Reviewer Summary:** This paper presents plume CO2 and d13C measurements of the 2021Tajogaite eruption.

The aspect that seems to be technically new in this paper is the direct measurement by UAS of CO2 to map CO2 plume concentrations to obtain the CO2 flux (as opposed to previous measurements where SO2 flux is measured and CO2 flux is calculated from the SO2 flux and a measurement of the CO2/SO2 ratio). I write "seems" because at a first reading, the paper is unclear about what is novel in the approach. That is my main criticism of the paper.

Overall, in my opinion, this paper deserves publication in AMT. However, I think it lacks clarity in terms of what is new in the present publication and what is not new (done by previous studies). For example, the authors start the second paragraph of the introduction with "We present of novel approach...". I think such a development should appear after the authors have reviewed what has been done previously by the studies on the subject to highlight what is missing in terms of methodology. Then if the approach presented in the authors has never been used before, the reader will rapidly capture how legitimate this novel study is. Because this is unclear, I went back to read previous papers on the subject to ask myself the following question: what's new? Rewriting should help in clarifying the paper.

**Response:**

We clarified and added to the contributions in the introduction.

"The main contributions of this work are that, for the first time, we estimate $CO_2$ flux using direct in-plume $CO_2$ measurements rather than using in-plume $CO_2/SO_2$ ratios combined with separately measured $SO_2$ emissions. The second major contribution is that we perform in-situ gas sample-return during a major volcanic eruption for carbon isotope measurements."

Another feature of the paper that is highly damaging to the understanding of the study is that the Methods section (section 4) is located at the end of the paper. To me, it should be located between the Background and the Results sections. That will greatly help clarity. When the reader gets to the results, there are many points of the methodology that the reader wants to know before integrating what has been obtained and what the results mean.

**Response:**

We moved the methods section to the proper location

Another point that can help clarity would be the presence of the map of Canary archipelago and the island where the study took place. The map in figure 2 is too restricted to help reading the Background section and also too unprecised : where is the lava vent? Are gas plumes different from lava vents?

**Response:**

We include a map of the archipelago and the location of the volcano as insert in Figure 3. The gas plumes originate from several lava vents. The lava vents are on the volcanic edifice and their locations are dynamic and changing frequently.

I did not get the reason why the flux in the abstract (1.19x10^6 to 2.80x10^7 t day-1 is different from the one quoted later in the text (line 93) : 2.6x10^4 to 5.4x10^4 t day-1, and in tables 2, 3 and 4. Are they from different periods etc. In the end, I am lost in all these numbers. If in the abstract these are the total flux over a given period, then this should be indicated.

**Response:**

We have made all the fluxes consistent throughout the paper and clarified when the data were collected.

Lines 80-85: I think the d13C value of modern air is not constant because there is a seasonal effect.

**Response:**

The reviewer is correct in that the $\delta^{13}C$ value of modern air is not constant. However, variations are quite small globally. For example $\delta13C$ values of  - 8.6‰ at Mauna Loa,  - 8.1‰ at Point Barrow, Northern Alaska, and  - 8.5‰ at La Jolla, California are reported [Keeling et al., 2005 and  http://scrippsco2.ucsd.edu/sites/default/files/graphics_gallery_images/c13_sta_records.png]. We state that we use our own sample collected at the ground locally and away from the volcano which was -8.0 ‰. We think that this is appropriate for our extrapolations, rather than using an assumed value for air, as it takes into account any instrumental uncertainties, i.e. all data were analyzed using the same instrument.

Lines 99-100: the mass balance is ultimately a little disturbing because mixing a source at 0 per mil and one at -5 per mil cannot explain a value above 0 (+0.1 per mil). Of course, with the quoted uncertainties, one can get to that but this leads to a strange reading.

**Response:**

The extrapolation using all of our data, shows that we obtain values of -1.4 to + 1.6‰. We state in the text that these values are consistent with those from phenocrysts in samples collected at the neighboring island of el Hierro. As detailed in response to point 3, we also state that the results are consistent with data collected on erupted phenocrysts from this Tajogaite eruption. We have removed the notion (in the discussion) that these values could be the result of mixing between mantle (-5) and carbonate (0) and focus on the consistency with results obtained by other workers.

---

## Referee Report (RR1)

**Second Review of "Drone CO₂ Measurements During the Tajogaite Volcanic Eruption" by Ericksen et al. (2024)**

The manuscript by Ericksen et al. (2024) applies Unpiloted Aerial System (UAS) platforms to measure carbon dioxide ($CO_2$) concentrations and carbon isotope ratios during the 2021 eruption of the Tajogaite Volcano in Spain. This study used a Dragonfly UAS outfit with systems for measuring $CO_2$ concentrations and carbon isotopic ratios for 10 transects through volcanic plumes during the eruption. Using measured $CO_2$ concentrations and winds, applying gaussian assumptions, led to emission rate estimates of $4.6\pm0.46 \times 10^3$ to $2.8\pm0.28 \times 10^4$ t day$^{-1}$ (4.6 to 28 kt day$^{-1}$). These emission rates are much more consistent compared to recent literature estimates compared to what was presented in the first version of the manuscript. Overall, the authors did a decent job in addressing my initial comments. The only major concern that remains is the author's minimal effort to estimate uncertainty in the $CO_2$ flux estimates. Please see my comment below. I think an improved uncertainty estimate, following other recent research cited below, would make this publication suitable for publication.

**Major Comments**

1. The uncertainty estimates of the $CO_2$ fluxes in this study are likely much too conservative. Many studies have shown that modeled winds (this study uses ERA5 model predicted wind speeds) are much larger than 10% (e.g., Nassar et al., 2017, 2021; Reuter et al., 2019; Johnson et al., 2020; Lin et al., 2023). Especially when you consider model wind speed and direction. Also, studies have shown it is not safe to assume a linear impact of wind speed on model prediction uncertainties (Nassar et al., 2017)? While wind speed/direction likely does have a majority impact on the overall uncertainty, it is not safe to neglect the other sources of uncertainty (e.g., measurement error, background concentration error, vertical distribution. etc.). It would be easy for this study to follow methods from recent research which quantify uncertainties from point-sources (.g., Nassar et al., 2017, 2021; Reuter et al., 2019; Johnson et al., 2020; Lin et al., 2023) to calculate more representative uncertainty values for this study.

**References**

Johnson, M. S., Schwandner, F. M., Potter, C. S., Nguyen, H. M., Bell, E., Nelson, R. R., et al. (2020). Carbon dioxide emissions during the 2018 Kilauea volcano eruption estimated using OCO-2 satellite retrievals. Geophysical Research Letters, 47, e2020GL090507. https://doi.org/10.1029/2020GL090507.

Lin, X., van der A, R., de Laat, J., Eskes, H., Chevallier, F., Ciais, P., Deng, Z., Geng, Y., Song, X., Ni, X., Huo, D., Dou, X., and Liu, Z.: Monitoring and quantifying CO2 emissions of isolated power plants from space, Atmos. Chem. Phys., 23, 6599–6611, https://doi.org/10.5194/acp-23-6599-2023, 2023.

Nassar, R., Hill, T. G., McLinden, C. A., Wunch, D., Jones, D., & Crisp, D. (2017). Quantifying CO2 emissions from individual power plants from space. Geophysical Research Letters, 44, 10,045–10,053. https://doi.org/10.1002/2017GL074702.

Nassar, R., Mastrogiacomo, J.-P., Bateman-Hemphill, W., McCracken, C., MacDonald, C. G., Hill, T., et al. (2021). Advances in quantifying power plant CO2 emissions with OCO-2. Remote Sensing of Environment, 264, 112579. https://doi.org/10.1016/j.rse.2021.112579.

Reuter, M., Buchwitz, M., Schneising, O., Krautwurst, S., O'Dell, C. W., Richter, A., et al. (2019). Towards monitoring localized CO2 emissions from space: Co-located regional CO2 and NO2 enhancements observed by the OCO-2 and S5P satellites. Atmospheric Chemistry and Physics, 19, 9371–9383. https://doi.org/10.5194/acp-19-9371-2019

---

## Author Response (AR2)

**Response to the Second Review of "Drone CO$_2$ Measurements During the Tajogaite Volcanic Eruption" by Ericksen et al. (2024)**

The manuscript by Ericksen et al. (2024) applies Unpiloted Aerial System (UAS) platforms to measure carbon dioxide (CO$_2$) concentrations and carbon isotope ratios during the 2021 eruption of the Tajogaite Volcano in Spain. This study used a Dragonfly UAS outfit with systems for measuring CO$_2$ concentrations and carbon isotopic ratios for 10 transects through volcanic plumes during the eruption. Using measured CO$_2$ concentrations and winds, applying gaussian assumptions, led to emission rate estimates of $4.6\pm0.46 \times 10^3$ to $2.8\pm0.28 \times 10^4$ t day$^{-1}$ (4.6 to 28 kt day$^{-1}$). These emission rates are much more consistent compared to recent literature estimates compared to what was presented in the first version of the manuscript. Overall, the authors did a decent job in addressing my initial comments. The only major concern that remains is the author's minimal effort to estimate uncertainty in the CO$_2$ flux estimates. Please see my comment below. I think an improved uncertainty estimate, following other recent research cited below, would make this publication suitable for publication.

**Response:**
We thank the reviewer for their second review of our manuscript. We have addressed the major concern raised by increasing our effort to estimate the uncertainty in the flux measurements. We have taken into account the works proposed by the reviewer and followed some of their methods. We show that the main uncertainty lies in the plume direction and we now state the modeled plume directions in Table 1. After accounting for all errors and summing them we obtain an error of $\pm$ 11.6%. This is quite close to our initial error estimate of $\pm$10%.

**Major Comments**

1. The uncertainty estimates of the CO$_2$ fluxes in this study are likely much too conservative. Many studies have shown that modeled winds (this study uses ERA5 model predicted wind speeds) are much larger than 10% (e.g., Nassar et al., 2017, 2021; Reuter et al., 2019; Johnson et al., 2020; Lin et al., 2023). Especially when you consider model wind speed and direction. Also, studies have shown it is not safe to assume a linear impact of wind speed on model prediction uncertainties (Nassar et al., 2017)? While wind speed/direction likely does have a majority impact on the overall uncertainty, it is not safe to neglect the other sources of uncertainty (e.g., measurement error, background concentration error, vertical distribution. etc.). It would be easy for this study to follow methods from recent research which quantify uncertainties from point-sources (.g., Nassar et al., 2017, 2021; Reuter et al., 2019; Johnson et al., 2020; Lin et al., 2023) to calculate more representative uncertainty values for this study.

**Response:**

We have re-evaluated our ERA5 output and have included the wind directions in Table 1. The range of wind-directions based on this model during our transect flights is $\pm15°$. This results in a flux estimate error of $\pm$ 3.4% based on our method to obtain linear distance of the transect: cos

(heading$_{UAS}$ – heading$_{wind}$), as described in the methods section. We have also included a 1% error of our $CO_2$ sensor measurement and a 1% error on the background measurement. Our estimated total error calculated by the root sum square method (following approaches of related work) results in an overall error of ±11.61% on our flux measurements for each transect.

We have described this in the revised version in particular at the end of section 2 (methods): "Uncertainty in the flux calculation is given by the following root sum of squares method which combines the uncertainties in wind velocity $\epsilon v$, wind direction $\epsilon d$ sensor error $\epsilon s$, and background $CO_2$ $\epsilon b$. $\epsilon$ is calculated similar to uncertainty calculation techniques described in Nassar et al. (2021); Lin et al. (2023); Nassar et al. (2017); Johnson et al. (2020)"

We also describe this in section 3.1 (plume transect wind measurements):
"The wind direction given by the ERA5 model yielded results ranging from 38◦ to 68◦ with an average of 53◦. These ranges contribute to the overall uncertainty $\epsilon_d$ "

and in section 4.1, the discussion of the $CO_2$ flux uncertainty:
"We used our wind estimates during the time of each flux calculation. This variation in wind velocity $\epsilon v$ is ± 11% which is calculated from the wind velocity range measured over the experiments (Table 1). The range of wind directions is ±15° from Table 1, which gives an error in the flux estimate based on $\epsilon d = 1-\cos(angle)$, thus ±3.40%. The SBA-5 documentation reports sensor error $\epsilon s$ is 1% in the range of $CO_2$ we measured. Finally, background ambient $CO_2$ $\epsilon b$ adds 1% to the uncertainty model which we calculated from the uncertainty in ambient $CO_2$ readings. Therefore, our estimated flux uncertainty given by the root sum of squares method is $\epsilon = \pm$ 11.61%."

**References**

Johnson, M. S., Schwandner, F. M., Potter, C. S., Nguyen, H. M., Bell, E., Nelson, R. R., et al. (2020). Carbon dioxide emissions during the 2018 Kilauea volcano eruption estimated using OCO-2 satellite retrievals. Geophysical Research Letters, 47, e2020GL090507. https://doi.org/10.1029/2020GL090507.

Lin, X., van der A, R., de Laat, J., Eskes, H., Chevallier, F., Ciais, P., Deng, Z., Geng, Y., Song, X., Ni, X., Huo, D., Dou, X., and Liu, Z.: Monitoring and quantifying CO2 emissions of isolated power plants from space, Atmos. Chem. Phys., 23, 6599–6611, https://doi.org/10.5194/acp-23-6599-2023, 2023.

Nassar, R., Hill, T. G., McLinden, C. A., Wunch, D., Jones, D., & Crisp, D. (2017). Quantifying $CO_2$ emissions from individual power plants from space. Geophysical Research Letters, 44, 10,045–10,053. https://doi.org/10.1002/2017GL074702.

Nassar, R., Mastrogiacomo, J.-P., Bateman-Hemphill, W., McCracken, C., MacDonald, C. G., Hill, T., et al. (2021). Advances in quantifying power plant $CO_2$ emissions with OCO-2. Remote Sensing of Environment, 264, 112579. https://doi.org/10.1016/j.rse.2021.112579.

Reuter, M., Buchwitz, M., Schneising, O., Krautwurst, S., O'Dell, C. W., Richter, A., et al. (2019). Towards monitoring localized $CO_2$ emissions from space: Co-located regional $CO_2$ and $NO_2$ enhancements observed by the OCO-2 and S5P satellites. Atmospheric Chemistry and Physics, 19, 9371–9383. https://doi.org/10.5194/acp-19-9371-2019